# K2S Challenge: From Undersampled K-Space to Automatic Segmentation

**DOI:** 10.3390/bioengineering10020267

**Published:** 2023-02-18

**Authors:** Aniket A. Tolpadi, Upasana Bharadwaj, Kenneth T. Gao, Rupsa Bhattacharjee, Felix G. Gassert, Johanna Luitjens, Paula Giesler, Jan Nikolas Morshuis, Paul Fischer, Matthias Hein, Christian F. Baumgartner, Artem Razumov, Dmitry Dylov, Quintin van Lohuizen, Stefan J. Fransen, Xiaoxia Zhang, Radhika Tibrewala, Hector Lise de Moura, Kangning Liu, Marcelo V. W. Zibetti, Ravinder Regatte, Sharmila Majumdar, Valentina Pedoia

**Affiliations:** 1Department of Bioengineering, University of California, Berkeley, CA 94720, USA; 2Department of Radiology and Biomedical Imaging, University of California San Francisco, San Francisco, CA 94158, USA; 3Department of Radiology, Klinikum Rechts der Isar, School of Medicine, Technical University of Munich, 81675 Munich, Germany; 4Department of Radiology, Klinikum Großhadern, Ludwig-Maximilians-Universität, 81377 Munich, Germany; 5Cluster of Excellence Machine Learning, University of Tübingen, 72076 Tübingen, Germany; 6Center for Computational and Data-Intensive Science and Engineering, Skolkovo Institute of Science and Technology, 121205 Moscow, Russia; 7Department of Radiology, University Medical Center Groningen, 9713 GZ Groningen, The Netherlands; 8Center for Advanced Imaging Innovation and Research, New York University Grossman School of Medicine, New York, NY 10016, USA

**Keywords:** image reconstruction, segmentation, multi-task learning, magnetic resonance imaging, musculoskeletal, deep learning, compressed sensing

## Abstract

Magnetic Resonance Imaging (MRI) offers strong soft tissue contrast but suffers from long acquisition times and requires tedious annotation from radiologists. Traditionally, these challenges have been addressed separately with reconstruction and image analysis algorithms. To see if performance could be improved by treating both as end-to-end, we hosted the K2S challenge, in which challenge participants segmented knee bones and cartilage from 8× undersampled k-space. We curated the 300-patient K2S dataset of multicoil raw k-space and radiologist quality-checked segmentations. 87 teams registered for the challenge and there were 12 submissions, varying in methodologies from serial reconstruction and segmentation to end-to-end networks to another that eschewed a reconstruction algorithm altogether. Four teams produced strong submissions, with the winner having a weighted Dice Similarity Coefficient of 0.910 ± 0.021 across knee bones and cartilage. Interestingly, there was no correlation between reconstruction and segmentation metrics. Further analysis showed the top four submissions were suitable for downstream biomarker analysis, largely preserving cartilage thicknesses and key bone shape features with respect to ground truth. K2S thus showed the value in considering reconstruction and image analysis as end-to-end tasks, as this leaves room for optimization while more realistically reflecting the long-term use case of tools being developed by the MR community.

## 1. Introduction

Magnetic Resonance Imaging (MRI) has emerged as one of the strongest medical imaging modalities for clinical use, offering exquisite soft tissue contrast for visualizing tissues such as ligaments, cartilage, and muscle [1,2]. Conventional MR sequences see the weighting of images in accordance with intrinsic MR parameters such as T1 and T2 and allow for suppression or saturation of signal from tissue types such as fat or fluid [3,4]. As such, MR images can be tailored for a given clinical context. Furthering this are recent developments of advanced sequences such as zero echo time (ZTE) and ultrashort echo time (UTE), which allow for high-resolution imaging of additional tissues such as tendons in musculoskeletal imaging [5,6,7,8]. MR has the added advantage of not exposing subjects to ionizing radiation compared to alternatives such as radiographs and computed tomography (CT). Despite these advantages, however, MR faces several challenges, including (1) long acquisition times and (2) the requirement of time-consuming and laborious radiologist annotation and interpretation of images to extract clinical meaning [9,10].

Fortunately, several tools have been developed to address these concerns. In the case of long MR scan times, acquisitions can be accelerated by sampling fewer points in k-space, the raw frequency-based domain in which MRI signals are obtained. This undersampling induces aliasing artifacts in resulting images that can be removed by image reconstruction algorithms. In recent years, considerable effort has been put into developing several families of reconstruction approaches: (1) compressed sensing (CS) algorithms iteratively reconstruct images by ensuring consistency with acquired k-space and imposing sparsity on the reconstructed image in an alternate domain [11,12,13,14]; (2) parallel imaging (PI) algorithms exploit the redundancy of using multiple coils to acquire the same imaging volume to reduce acquisition times at the expense of signal-to-noise ratio (SNR) [15,16,17]; (3) deep learning (DL) approaches use complex, nonlinear models to impute full-length acquisition images from aliased images and/or undersampled k-space [18,19]. Other approaches growing in popularity include magnetic resonance fingerprinting (MRF) and low-rank and sparse modeling approaches [20,21,22]. On the other hand, a host of DL tools have emerged to automate mundane MR image-processing tasks. For instance, the introduction of the U-Net in 2015 seeded major advances in medical image segmentation from limited data, paving the way for more complicated architectures that have been applied for accurate lumbar spine and knee cartilage segmentation, among others [23,24,25,26]. Yet other DL applications include automatic assessments of cartilage thickness, staging of anterior cruciate ligament injury severity, diagnosis of lumbar spine anomalies, and analysis of bone shape [27,28,29,30].

This body of work unquestionably reflects substantial advances made by the MR research community. But it is noteworthy that with extremely few exceptions, the challenges of long acquisition times and image analysis have been treated as separate entities [31]. The long-term vision, however, would be a software package that addresses these challenges simultaneously, raising a niche for optimization. Namely, image analysis algorithms are designed for full-length acquisition of MR image inputs, but there is no guarantee and little investigation that they would perform similarly well on reconstructed images from the accelerated acquisition. On the other hand, image reconstruction algorithms are overwhelmingly optimized for metrics such as normalized root mean square error (nRMSE), peak signal-to-noise ratio (PSNR), and structural similarity index (SSIM), which correlate to the perceptual quality of a reconstructed image [32,33,34]. In other words, reconstruction algorithm image outputs are optimized for visually appealing images and radiologist interpretation, but what if their outputs were instead intended as features for subsequent image analysis pipeline input? Could the features required for accurate radiologist readings with respect to ground truth differ from those required for DL image analysis pipeline input to yield strong performance? More generally, if image reconstruction and annotation are viewed as end-to-end rather than serial tasks, is it possible to attain stronger image analysis performance?

To answer these questions, we hosted the K2S challenge at the 25th International Conference on Medical Image Computing and Computer-Assisted Intervention (MICCAI) in Singapore. Previously, challenges and/or the releases of large datasets have spurred major advances in the MR research community. The release of the Osteoarthritis Initiative (OAI) and Multicenter Osteoarthritis Study (MOST) precipitated substantial advances in understanding osteoarthritis, total knee replacement, and knee pain, among others [35,36,37,38,39]. On the other hand, the fastMRI challenge was crucial in (1) making image reconstruction more accessible to the MR research community by releasing large datasets including raw k-space, and (2) seeding major advances in reconstruction research such as popularizing unrolled DL architectures [40,41,42]. Our objective was to fill a similar niche in the end-to-end reconstruction and image analysis space. As such, we curated the K2S dataset, which consists of 300 patients that underwent 3D fat-suppressed knee scans, for each of whom k-space data and radiologist-approved 6-compartment tissue segmentations were released. The use of Fourier-transformed DICOM images as k-space would be problematic, not maintaining consistency with the multicoil nature of most MR acquisitions and the numerous post-processing steps to convert raw multicoil k-space into DICOM images, while also likely overstating performance [43]; importantly, our released dataset thus was of raw multicoil k-space data. Challenge participants were to train algorithms that segmented knee bones and cartilage from 8× undersampled acquisitions. Winners were selected using a weighted dice similarity coefficient (DSC) that assessed the accuracy of resulting segmentations, but additional analyses were conducted to assess segmentation quality, determine if strong image reconstruction was a prerequisite for strong segmentation, and gage the suitability of submitted segmentations for biomarker analysis [44].

In short, the contributions of the K2S challenge and this paper are as follows:Reframing image reconstruction and annotation as end-to-end tasks for an eventual clinical workflow rather than sequential steps.Curating a large dataset (n = 300) with 3D raw k-space data and tissue segmentations to allow training of segmentation algorithms directly from undersampled k-space, and whatever additional research objectives may emerge from having raw k-space and segmentations in the same dataset.Investigating whether strong image reconstructions are a prerequisite for strong tissue segmentations.Assessing if segmentation algorithms trained from 8× undersampled data are suitable for biomarker analysis.

## 2. Materials and Methods

### 2.1. Challenge

K2S challenge participants were responsible for predicting 6-class knee tissue segmentations (femur, tibia, patella, femoral cartilage, tibial cartilage, and patellar cartilage) from 8× undersampled k-space data. An overview of the steps involved in dataset curation, the challenge objective, and the timeline can be viewed in Figure 1, with details on all steps and evaluation criteria described below.

### 2.2. Dataset

#### 2.2.1. Subject Eligibility and Sequence Information

Subjects at the UCSF Orthopedic Institute between 14 June 2021 and 21 June 2022 were scanned with an imaging protocol that included 3D fat-suppressed CUBE acquisitions (n = 816). There were no exclusion criteria placed on patients for inclusion in the eventual K2S dataset, and patients were scanned in accordance with all pertinent guidelines, including approval from the UCSF Institutional Review Board (Human Research Protection Program), obtaining informed consent from all study participants. The 3D fat-suppressed CUBE sequence was selected for K2S, as 3D sequences have higher SNR compared to 2D imaging, allowing for higher resolution acquisitions that can be reformatted into multiple planes for subsequent research objectives. Scans were performed on a GE Discovery MR750 3T Scanner using an 18-channel knee transmit/receive coil. The full-length acquisition time of the sequence was 4 min and 58 s. Complete acquisition parameters are listed in Table 1.

#### 2.2.2. Extraction of ARC-Reconstructed Multicoil Raw k-Space Data

An in-house pipeline was developed replicating all post-processing steps done on an MR scanner to go from raw k-space data to DICOM images viewed by clinicians for diagnostic decisions. To the best of the authors’ knowledge, no centralized resource is available describing all these steps, which can make it difficult for those interested in reconstruction to familiarize themselves with the process before model development. The authors thus saw value in describing these steps, shown schematically in Figure 2, with examples of pipeline intermediates at several steps in Figure 3. Unless otherwise specified, all post-processing steps were implemented using functions in GE Orchestra 1.10.

##### k-Space Post-Processing

Some sequences may leverage PI techniques (such as ARC or GeneRalized Autocalibrating Partial Parallel Acquisition (GRAPPA)) to acquire fewer lines within k-space, instead exploiting already acquired data across multiple coils to mitigate aliasing artifacts at the expense of SNR [17,45]. This was the case for our sequence; consequently, the first step in post-processing raw multicoil k-space data was applying ARC to impute unacquired k-space lines. Subsequently, Fermi filtration was applied: given MR images are often zero-padded in k-space, ringing artifacts can emerge from the sharp boundary in k-space between nonzero and zero points [46]. A Fermi filter smooths this boundary, reducing ringing artifacts at the expense of sharpness in the reconstructed image. A custom Fermi filtration function was used, using the Fermi filtration radius and width parameters extracted from raw sequence metadata. After Fermi filtration, k-space was zero-padded to the intended image dimensions (in our case, from 256 × 256 × 200 to 512 × 512 × 200), completing k-space post-processing. All k-space post-processing was on multicoil data.

##### Image Space Post-Processing

Post-processed k-space was 3D inverse Fourier transformed to image space for each of the 18 coils and coil-combined to yield a single-coil image. The most basic means of coil combination is root sum-of-squares, but GE provides another method based on Array coil Spatial Sensitivity Encoding (ASSET), which leverages sensitivity maps in a PI-inspired technique to do coil combination [47]. Magnitude images were then calculated, after which GE’s Phased array Uniformity Enhancement (PURE) was used to perform surface coil intensity correction [48,49]. This was followed by GRADWARP, which warps images to correct for inhomogeneities in gradient coils [50]. A final step in post-processing was correcting image orientation and scaling pixel values, yielding DICOM images used by clinicians for diagnostic purposes.

In the context of segmenting undersampled images, one complication emerges: in GRADWARP, the MR image is warped such that it no longer corresponds to k-space. As such, the post-processing pipeline intermediate prior to GRADWARP must be segmented, or the GRADWARP function must be integrated into model training itself while segmenting DICOM images. Due to the difficulties of implementing the latter (backpropagating through GRADWARP would not be trivial), our solution was the former.

#### 2.2.3. Ground Truth Segmentation Generation

Ground truth knee cartilage and bone segmentations were generated by separate DL pipelines and post-processing techniques, each trained with a radiologist in the loop.

##### Cartilage Segmentation Pipeline

480 3D fat-suppressed CUBE sequences were acquired across three sites (UCSF, San Francisco, CA, USA; Hospital for Special Surgery, New York, NY, USA; Mayo Clinic, Rochester, MN, USA) with similar acquisition parameters to the 3D fat-suppressed CUBE sequences ultimately used in K2S. These volumes were manually segmented by readers trained by a senior radiologist with over 25 years of experience, split 400/80 into training and validation, and used to train a 3D V-Net for multiclass cartilage segmentation [51,52]. This initial pipeline was inferred on 20 3D fat-suppressed CUBE sequences from the UCSF Orthopedic Institute with K2S acquisition parameters, but on volumes acquired prior to the eligibility window for K2S inclusion. The 20 inferred segmentations were manually corrected and quality checked (QC) by an intern under radiologist supervision. 15 of the 20 cases were used to fine-tune the pipeline in a second training, seeing convergence reached after 5 epochs, and the remaining 5 cases were used to select final model parameters.

After the second training, the V-Net was inferred on all 816 cases eligible for K2S. The following post-processing steps were selected and applied under radiologist supervision: 3D morphological opening, 3D connected components analysis (preserving the largest femoral and patellar and the 2 largest tibial cartilage components), and 2D sagittal connected components analysis (preserving all connecting components larger than 150 pixels).

##### Bone Segmentation Pipeline

40 3D fat-suppressed CUBE sequences acquired at the UCSF Orthopedic Institute prior to the eligibility window for K2S inclusion were manually segmented by a trained reader for bone, tibia, and patella. These cases were used to train a baseline 3D U-Net for a binary bone segmentation model. An additional 15 cases acquired using the K2S acquisition parameters were also manually segmented by three radiologists with three (J.L.), three (P.G.), and four (F.G.) years of experience. The trained baseline model was inferred from these cases, which were used for model fine-tuning.

The fine-tuned U-Net was inferred on the 816 cases with the following post-processing steps, applied under radiologist supervision: filling holes, morphological opening, and connected components analysis (preserving all connecting components larger than 1000 voxels and with centroids in central 50% of slices). Finally, the sizes of connected components were used to extract bone labels (femur, tibia, and patella).

#### 2.2.4. Selection of Cases for K2S Dataset

Of the 816 potential cases, the target was selecting 300, with the intent of maintaining sufficient cases for training reconstruction and segmentation models, maintaining some variety of anomalies in included cases, and ensuring a reasonable memory footprint given computational constraints. Radiologists with three (J.L.) and four (F.G.) years of experience developed 5-point LIKERT scales to assess segmentation quality: (1) unusable; (2) poor, with some mislabeling of bones or cartilage; (3) useable, with some major issues, but correct labeling of bone or cartilage; (4) good, with some minor but acceptable issues; (5) (near) human-like. Examples of cartilage segmentation LIKERT scores for the 5 classes are seen in Figure 4, and for bone in Figure 5. Segmentation LIKERT scores were calculated for bone and cartilage from videos of the segmentations that cycled through all sagittal slices. Cases with acceptable segmentation quality for both cartilage and bone were selected as the K2S dataset. Cartilage LIKERT scores for K2S were as follows: 5:14; 4:175; 3:110; 1:1. Bone LIKERT scores were as follows: 5:112; 4:179; 3:9.

#### 2.2.5. Final K2S Dataset Characteristics

The K2S training dataset (n = 300) had the following demographic characteristics: age of 44.3 ± 13.9 years, weight of 75.6 ± 14.9 kg, 160/140 male to female. The test dataset followed the same described steps (n = 50): age of 44.5 ± 14.4 years, weight of 70.5 ± 16.6 kg, 26/24 male to female (all mean ± standard deviation).

The training dataset included the following: multicoil ARC-reconstructed k-space and multiclass segmentation for each patient (n = 300), 8× center-weighted Poisson undersampling mask with a fully sampled central 5% square of k-space in k_y_-k_z_, and a file detailing the quality of the segmentations and any radiologist notes associated with each patient. The released test dataset was solely the 8× undersampled multicoil ARC-reconstructed k-space.

### 2.3. Evaluation Process

Submissions were evaluated using a weighted sum of DSC. Namely, DSC was calculated in each of the 6 tissue compartments, and combined as follows into a weighted DSC that assigned each compartment a weight inversely proportional to the size of the tissue compartment:(1)$$Weighted\;DSC=\frac{\sum_{t}\frac{DSC_{t}}{n_{t}}}{\sum_{t}\frac{1}{n_{t}}}$$

In equation (1) [44], *t* refers to the tissue compartment, *DSC_t_* refers to the DSC within that tissue compartment, and *n_t_* is the number of pixels in the ground truth segmentation for tissue *t*.

### 2.4. Timeline

15 April 2022: Training dataset release30 April 2022: Participant registration close27 June 2022: Release of code used to evaluate submissions6 July 2022: Test dataset release21 July 2022: Submission deadline28 July 2022: Invitation of top 4 teams for in-person presentations18 September 2022: In-person workshop at MICCAI 2022, winners announced

All told, 87 teams registered for the K2S challenge from 19 countries, and 12 teams made submissions for the challenge.

### 2.5. Overview of Top Submission Methodologies

#### 2.5.1. K-nirsh (University of Tübingen, Tübingen, Germany)

K-nirsh’s submission involved two cascaded nnUNet architectures, a first for reconstruction and a second for segmentation [53]. Multicoil k-space was inverse Fourier transformed and coil-combined using root sum-of-squares coil combination, yielding coil-combined 8× undersampled images. An initial nnUNet was pretrained to predict fully-sampled coil-combined images from 8× undersampled coil-combined inputs using a mean square error (MSE) loss. A second nnUNet was pretrained to predict multiclass cartilage and bone segmentations from a 2-channel input (8× undersampled coil-combined image and fully sampled coil-combined image), using DSC segmentation loss. After pretraining, these models were trained end-to-end, with the initial nnUNet regression output replacing the fully sampled coil-combined image as input for the second segmentation nnUNet. The model was fine-tuned for over 1000 epochs on NVIDIA V100 GPUs, using only the segmentation loss and implementing a weight scheduler that linearly increased small class weighting (cartilage). The weighted DSC loss used to evaluate the challenge submission was used as a validation loss and the best model according to this metric was chosen for the challenge submission. The output of the first nnUNet was considered the reconstruction output of this pipeline, whereas the output of the second nnUNet was the segmentation submission.

#### 2.5.2. UglyBarnacle (Skolkovo Institute of Science and Technology, Moscow, Russia)

UglyBarnacle’s submission differed from other top submission methodologies by leveraging CS as opposed to DL for reconstruction. An initial reconstruction pipeline accepted as input the 18-channel, 256 × 256 × 200 8× undersampled k-space array, performing a CS reconstruction with a combined L_1_-wavelet and total variation (TV) regularization function, imposing 3 times the weight on TV as opposed to L_1_-wavelet. The CS reconstruction was solved as an optimization problem: the goal was to find the undersampled part of the k-space that minimized the target value function (weighted sum of L_1_-wavelet and TV of volumetric image). The optimization problem was solved using the Adam optimization algorithm over 50 iterations for each scan. Reconstructed images were fed to an architecture similar to V-Net for tissue segmentation. The segmentation network was implemented in 3D, with the following feature map depths at V-Net stages: 16, 32, 64, 128, 256. Max-pooling was used to compress the representation of feature maps in the encoder, and upsampling to increase resolution in the decoder, with skip connections transferring information from the encoder to corresponding parts of the decoder. The network output was fed through two final convolutions (one with a feature map depth of 7 and the last with a depth of 1) to yield predicted segmentations.

#### 2.5.3. FastMRI-AI (University Medical Center Groningen, Groningen, The Netherlands)

As with K-nirsh, k-space was zero-padded to 512 × 512 along k_x_ and k_y_, inverse Fourier transformed, and root sum-of-squares combined, yielding coil-combined 8× undersampled image space. Unlike other top submissions, FastMRI-AI did not implement a reconstruction framework, choosing instead to directly segment the undersampled image; the root sum-of-squares coil combined images were thus considered the reconstruction outputs for this approach in subsequent analysis. A 3D U-Net featuring a squeeze and excite attention layer was trained on 160 × 160 × 48 patches, selected with stride 51 × 51 × 16, yielding around 27 predictions per voxel [54]. Networks were trained with weighted DSC loss, giving twice the weight to cartilage afforded to bone and background. Predictions were post-processed with simulated extended image boundaries by mirror padding, self-ensembling for overlapping sliding window prediction, and connected component analysis for each class, removing objects that were less than 60% the size of the largest object in the given class.

#### 2.5.4. NYU-Knee AI (New York University Grossman School of Medicine, New York, USA)

NYU-Knee AI trained multiple components individually: a Variational Network (VN) for image reconstruction, followed by an ensemble of 2D U-Nets to predict tissue segmentations [55,56,57,58,59]. For reconstruction, eSPIRIT was used to calculate coil sensitivity maps for undersampled and ground truth data using the central 24 × 24 region in k-space [60]. Zero-filled k-space was then fed through a VN for K = 10 iterations, at each iteration using calculated coil sensitivity maps and acquired k-space to ensure data consistency with intermediate reconstructed images, while also feeding iteration outputs through a convolutional, ReLU, and transpose convolutional layer to encourage recovery of details lost from undersampling. The VN was trained with an MSE loss function between the 256 × 256 ground truth and the reconstructed coil-combined images for 200 epochs. VN outputs were fed to 2D U-Nets, predicting 256 × 256 segmentations that were upsampled and convolved to the intended 512 × 512 output resolution. Multiple networks were trained with either focal loss, cross-entropy loss, or a hybrid of both for 300 epochs; an internal validation set was used to choose the best-performing network for each of the 6 tissue classes, ultimately using 3 focal loss networks and 1 weighted cross entropy loss network in the final submission.

### 2.6. Further Analysis of Submissions

#### 2.6.1. Intermediate Pipeline Reconstruction Performance

The objective of the challenge was segmenting bones and cartilage, and no part of the evaluation criteria nor any communication between organizers and challenge participants prior to submissions discussed a requirement for reconstruction submissions. However, at some level, each of the top-performing pipelines fed some image (either directly undersampled for FastMRI-AI, or after reconstruction for the other 3 top submissions) through a segmentation pipeline. As such, it was instructive to see how reconstruction metrics of images fed to segmentation pipelines compared to segmentation metrics. Challenge organizers thus requested the top four teams provide intermediate reconstruction outputs for the test set. Using these images, standard reconstruction metrics were calculated: nRMSE, PSNR, and SSIM.

#### 2.6.2. Comparison of Reconstruction and Segmentation Performance

In addition to the visual comparison of reconstructions and segmentations, Pearson’s r was calculated between weighted DSC and nRMSE, PSNR, and SSIM for each of the top 4 teams [61]. Given the wide variety of approaches used by the teams, these experiments investigated a correlation between reconstruction and segmentation performance.

#### 2.6.3. Biomarker Analysis: Cartilage Thickness

Previously developed tools were used to calculate cartilage thicknesses for ground truth and submissions [27]. Briefly, Euclidean distance transforms on each cartilage compartment of each patient were used to generate skeletonizations. The skeletonizations were sampled and distances from skeletonized points to cartilage surfaces were calculated for each compartment and each patient. Skeleton-to-surface distances were averaged across a cartilage compartment for a given patient to obtain mean cartilage thickness measurements, which were then compared between ground truth and each of the submissions in Bland-Altman and correlation plots. Pearson correlation coefficients were calculated for each submission to assess the correlation of submitted cartilage thicknesses to ground truth, as a proxy for assessing the suitability of submissions for biomarker analysis.

To visualize cartilage thickness maps, voxel-based segmentations were converted into triangulated meshes using a Marching Cubes algorithm, and cartilage thickness maps were projected onto bones for select cases [39]. Maps were then compared for a qualitative assessment of regions best and most poorly preserved by sample submissions.

#### 2.6.4. Biomarker Analysis: Bone Shape

To analyze the bone shape, previously developed tools again were applied [30]. Triangulated meshes of each bone of the ground truth segmentations were generated using a Marching Cubes algorithm, after which Euclidean coordinates of each point in the mesh were flattened into a 1D vector for each test set case. Principal component analysis (PCA) was used to reduce the dimensionality of these vectors, preserving the top 5 PCs, which constituted bone shape features. Statistical parameterization was used to extract the mean and standard deviation of each PC. For visualization purposes, mean +3 standard deviations (s.d.) and mean −3 s.d. bone shapes were generated for each PC, with qualitative interpretations of the features varying most with each PC being described (i.e., volume). Segmentations of each submission were similarly transformed into 1D Euclidean coordinate vectors and projected into the PC space generated from the ground truth. Correlations between submissions and ground truth along these shape features were calculated for each.

## 3. Results

12 submissions were received for the K2S, for which weighted DSC was calculated across the test set as described in Equation (1). The top four submissions by weighted DSC were analyzed further, with results discussed below.

### 3.1. Segmentation Metrics

Segmentation results are shown in Table 2, stratified by tissue compartment but also showing the weighted DSC that determined challenge winners. K-nirsh delivered strong segmentation performance in each tissue compartment, closely rivaling ground truth, and interestingly did so from intermediate reconstruction outputs exhibiting poor reconstruction metrics. FastMRI-AI also yielded high-quality segmentations despite not implementing any reconstruction framework. Overarchingly, segmentation performance for all four pipelines was strong, given that severely aliased images served as model input. To differentiate between the top two submissions, which showed similar weighted DSC, a paired t-test was run to assess for significant difference in performance: K-nirsh performance indeed was significantly better than UglyBarnacle, even after adjusting for Bonferroni correction (n = 50, α = 0.05).

### 3.2. Reconstruction Metrics

Example sagittal slices of intermediate pipeline reconstruction outputs are shown in Figure 6, with corresponding reconstruction metrics. NYU-Knee AI and particularly UglyBarnacle produced intermediate reconstruction outputs with strong fidelity to ground truth, recovering fine details lost to aliasing. On the other hand, K-nirsh yielded an image with more distinct tissue boundaries, but with noise and pixel intensity distributions that clearly differed from the ground truth. Complete metrics of reconstruction performance are shown in Table 3.

### 3.3. Comparison of Reconstruction and Segmentation Performance

Example slices of predicted segmentations, overlaid on intermediate reconstruction outputs, are shown for all four teams alongside ground truth in Figure 7. For each reconstruction metric, and for each of the top 4 performing pipelines, weighted DSC was plotted against the reconstruction metric in Figure 8, with Pearson’s correlation coefficients being calculated for each pair. The highest correlation coefficient in this study was between nRMSE and weighted DSC for the NYU-Knee AI submission, at 0.284, with all other correlation coefficients being substantially lower. This indicates that, at the absolute best, there was a weak correlation between segmentation and reconstruction metrics, and in most cases, there was a negligible or even slightly negative correlation between the two.

### 3.4. Biomarker Analysis: Cartilage Thickness

Example femoral cartilage thickness maps projected onto the femur are shown in Figure 9, with corresponding femoral cartilage segmentation DSCs. These results elucidate added complexity: while FastMRI-AI and NYU-Knee AI lagged K-nirsh and UglyBarnacle in weighted DSC, they did a better job preserving certain thick and thin cartilage regions. Qualitatively, however, these maps show K-nirsh, UglyBarnacle, and FastMRI-AI perform especially well in reconstructing cartilage thicknesses; Bland-Altman plots in Figure 10 confirm these results, showing cartilage thicknesses across all three compartments were predicted with minimal bias and strong fidelity to ground truth by these three teams. Interestingly, bias in retaining femoral cartilage thicknesses decreased with larger ground truth cartilage thicknesses, regardless of submission. More granularly, while fastMRI-AI slightly overestimated patellar cartilage thicknesses, they also reflected the least bias in maintaining femoral cartilage thickness, showing some discordance between weighted DSC and downstream biomarker analysis. Contrarily, thicknesses were overestimated by NYU-Knee AI, particularly in tibiofemoral regions. In comparing the top two challenge finishers, K-nirsh and UglyBarnacle, biases in predicted cartilage thicknesses were slightly lower for UglyBarnacle in femoral and tibial cartilage, and slightly higher in patellar cartilage (UglyBarnacle: femoral: 0.088 ± 0.07, tibial: 0.036 ± 0.09, patellar: 0.114 ± 0.13; K-nirsh: femoral: 0.096 ± 0.08, tibial: 0.049 ± 0.09, patellar: 0.097 ± 0.11; all in units of mm, mean ± 1 s.d.). However, paired t-tests showed none of these differences were significant even after Bonferroni correction (n = 50, α = 0.05).

Correlation plots in Figure 10 showed K-nirsh, UglyBarnacle, and NYU-Knee AI yielded high Pearson correlation coefficients with respect to ground truth, indicating high-quality segmentations. Interestingly, UglyBarnacle showed a slightly higher correlation to ground truth cartilage thickness in tibiofemoral cartilage than K-nirsh, despite lower DSCs in both tissues. Visually, FastMRI-AI also appeared to show a strong correlation between predicted and ground truth cartilage thickness, although poor prediction in one case appeared to severely degrade the correlation coefficient.

### 3.5. Biomarker Analysis: Bone Shape

Statistical shape modeling identified 5 femoral shape features most contributing to variation within the test set, as illustrated in Figure 11: femoral volume, medial wall incline slope, condylar posterior protrusion, intercondylar notch width, and width-to-height ratio. Similar features were identified for the patella and tibia, and the correlation between submitted bone shapes and ground truth was calculated for the top PCs (and thus, top shape features) for each submission. Those correlation coefficients are shown in Figure 12: while each of the top four submissions performed best in at least one of the 15 shape features across the 3 bones, generally K-nirsh had the strongest performance among the teams in the femur, while NYU-Knee AI did best within the tibia and patella. Correlations for all teams were moderate to strong for many of the shape features.

## 4. Discussion

In this work, we describe the K2S challenge, which aims to reframe image reconstruction and image analysis as end-to-end rather than serial tasks, opening room for optimization. We curated the K2S dataset of 300 patients that had undergone 3D fat-suppressed knee MRI acquisitions, each with 3D raw k-space and bone and cartilage segmentations, challenging participants to segment the tissues directly from 8× undersampled k-space. A variety of solutions were submitted for the challenge. Some, like NYU-Knee AI and UglyBarnacle, spent considerable time optimizing reconstruction networks, leveraging VN and CS frameworks to attain high-quality reconstructions that served as inputs for standard segmentation networks. Interestingly, FastMRI-AI did not pursue a reconstruction network at all, choosing exclusively to optimize the segmentation network and develop unique postprocessing techniques, attaining very competitive results. K-nirsh, on the other hand, pretrained separate reconstruction and segmentation networks, performing end-to-end optimization of both for weighted DSC. The end-to-end optimization made this the only approach that implicitly optimized reconstruction outputs for segmentation inputs, possibly playing a role in their top finish within the challenge. All told, however, all top submissions produced high-quality segmentations in knee cartilage and bone, maintaining accuracy with respect to ground truth despite working originally from 8× undersampled multicoil k-space.

Beyond strong DSC metrics, predicted segmentations from all top submissions produced cartilage thickness maps that either maintained minimal bias or strong correlation to ground truth cartilage thicknesses. Statistical shape modeling generated five features that captured the most variance in bone shape for each of the patella, tibia, and femur. Each of the top submissions was most correlated to ground truth along at least one of the features, with moderate to good correlations seen in many, while K-nirsh and NYU-Knee AI generally showed the best performance in retaining bone shape. As such, for both bone and cartilage, all top submissions yielded cartilage and bone segmentations that to varying degrees were suitable for subsequent biomarker analysis. An added observation was that downstream biomarker performance did not always correspond with segmentation metrics: for instance, NYU-Knee AI delivered among the best correlations between predicted and ground truth bone shape features despite obtaining the poorest weighted DSC among the top 4 submissions, with segmentations that often appeared slightly dilated compared to ground truth but preserved shape. Likewise, UglyBarnacle slightly outperformed K-nirsh in correlations between tibiofemoral cartilage thicknesses and ground truth despite slightly poorer weighted DSCs, but its slightly reduced bias was not statistically significant. This accentuates the complexity of segmentation as an image analysis task: there is no all-encompassing, perfect metric to quantify segmentation quality.

A noteworthy finding from this challenge was that strong reconstruction performance was not a prerequisite for strong segmentation performance. K-nirsh had by far the poorest metrics of the submitted pipeline reconstruction intermediates, poorer even than the root sum-of-squares coil-combined 8× undersampled images that FastMRI-AI used as pipeline inputs. Despite this, K-nirsh yielded the strongest segmentation performance; visual inspection of K-nirsh reconstructions reveals sharp images that enhance contrast at boundaries between different tissues such as cartilage/bone boundaries, yielding an image that is perhaps easier to segment than ground truth. This demonstrates that ideal features for radiologist interpretation of an MR image can differ from those optimal for processing by an image analysis algorithm. That FastMRI-AI showed competitive segmentation performance despite directly segmenting undersampled images is a testament to this. Furthering this, was there essentially no correlation between reconstruction and segmentation metrics for any of the top submissions on a per-patient basis. There is therefore room for optimizing image analysis algorithms when trained end-to-end with reconstruction algorithms instead of training separate algorithms and inferring serially. It is important to note that segmentation performance from undersampled k-space depends not only on the segmentation algorithm but also on the undersampling pattern, which was fixed in this challenge. More complicated joint optimization of segmentation and undersampling can further improve end-to-end MRI reconstruction and image analysis outcomes [62,63].

Apart from the specific challenge, the curation and release of the K2S dataset marks an important initiative that can seed advances in both reconstruction and image analysis algorithm development. To our knowledge, this is the largest released dataset that pairs raw k-space data with tissue segmentations (n = 300 patients, each with an 18-coil, 200-slice k-space). While a dataset of this size is more than sufficient for training most reconstruction algorithms, image annotation algorithms generally require considerably larger datasets to sufficiently represent rare anomaly classes. Our hope is that the release of this dataset can allow research groups to investigate objectives such as ROI-specific image reconstruction, end-to-end reconstruction and segmentation, and more generally end-to-end reconstruction and image analysis tasks.

This challenge had some limitations. First off, the k-space provided to challenge participants had undergone R = 4 ARC, and thus does not reflect the full-length acquisition k-space that would ordinarily be undersampled. Given that the full 3D fat-suppressed CUBE sequence without ARC would require nearly 20 min for acquisition, this compromise was made to make curate a larger dataset suitable for algorithm development. Additionally, while substantial work was done by challenge organizers and radiologists (J.L. and F.G.) in inspecting segmentation quality, bone and cartilage segmentations ultimately were model generated, and were not the gold-standard manual annotations that are desired for training models. It is therefore more accurate to describe the challenge objective as achieving on 8× undersampled data the same segmentation performance seen on fully sampled data, albeit the latter was carefully monitored and quality checked by radiologists. This tradeoff was taken to obtain a substantially larger dataset than would have been possible if exclusively using manual segmentations. We would expect these findings to hold on a dataset with purely manual segmentations but confirming so would require inferring trained models on such a dataset. Furthermore, this challenge provided a fixed undersampling pattern: a center-weighted Poisson pattern with a fully-sampled center. This undersampling pattern was selected such that potential challenge solutions would not be biased towards or against a given reconstruction backbone (i.e., compressed sensing, deep learning), but there conceivably would be room for further optimization of segmentations with respect to the undersampling pattern. Additionally, since all submissions were trained and tested on a fixed undersampling pattern, the robustness of solutions to other R = 8 undersampling patterns was not assessed and is an important research objective for the reconstruction community to pursue. Lastly, there is no perfect solution to the gradient inhomogeneity correction step (GRADWARP) in the standard processing pipeline of raw scanner data. Once applied, correspondence between k-space and image space is lost, meaning ordinary DICOM image segmentations would not match k-space. In the K2S dataset, segmentations were provided on images prior to GRADWARP application, meaning that gradient coil inhomogeneities manifested themselves into segmentations. Due to the difficulty in backpropagating through GRADWARP, this was viewed as the easier choice for pipeline development, with the understanding that resulting segmentations could be processed by GRADWARP to perform necessary corrections. Nonetheless, this is an unavoidable limitation that must be discussed at greater length for this and other datasets that may be released pairing k-space and tissue segmentations.

In conclusion, the K2S challenge curated a landmark dataset, tasking participants with segmenting bone and cartilage from 8× undersampled knee MRI images. Through it, the top four teams produced submissions that yielded high-quality segmentations, showing highly varied methodologies and very strong performance that was suitable for downstream biomarker analysis in cartilage thickness and bone shape assessments. Through the submissions of two teams with unconventional approaches—K-nirsh and FastMRI-AI—we clearly see that features required for radiologist annotation differ from those required for DL model input, there is room for image analysis pipeline optimization when trained end-to-end with reconstruction, and strong reconstruction is not a prerequisite for strong segmentation. These findings can motivate similar efforts for end-to-end optimization of image analysis and reconstruction tasks, not only for segmentation, but for anomaly detection, prognosis prediction, bone shape assessment, and others.

## Figures and Tables

**Figure 1 bioengineering-10-00267-f001:**
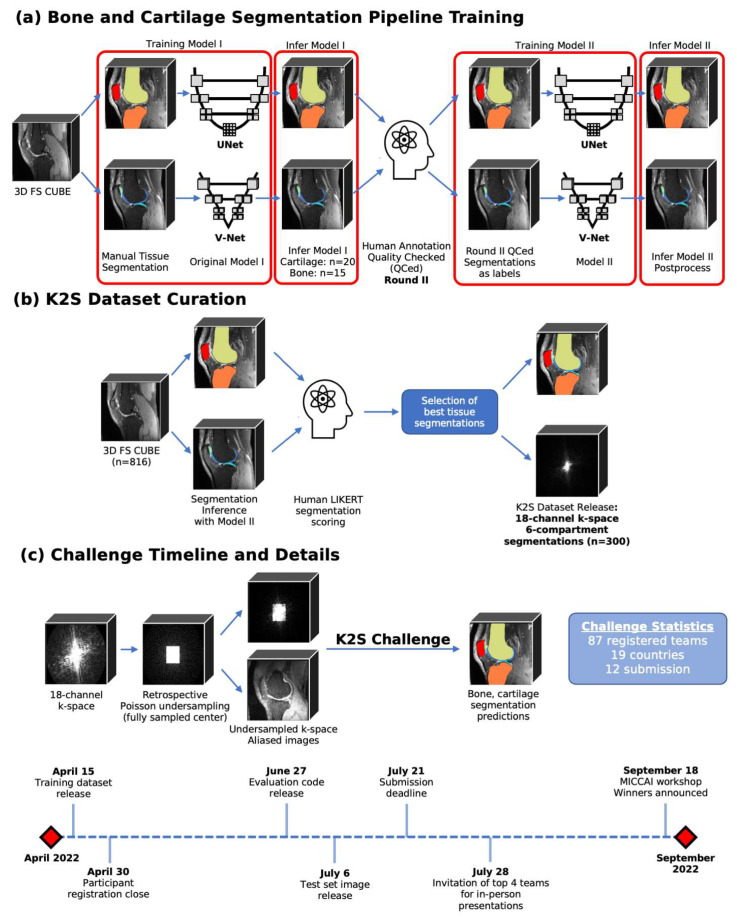
Overview of steps involved in human-in-the-loop training of models to generate ground truth bone and cartilage segmentations, and the process for radiologist approval of final 300 segmentations to be included in K2S dataset. The K2S challenge was for participants to segment knee bones and cartilage from 8× undersampled k-space, with the training set released on 15 April, the test set released on 6 July, and the submission deadline on 21 July.

**Figure 2 bioengineering-10-00267-f002:**
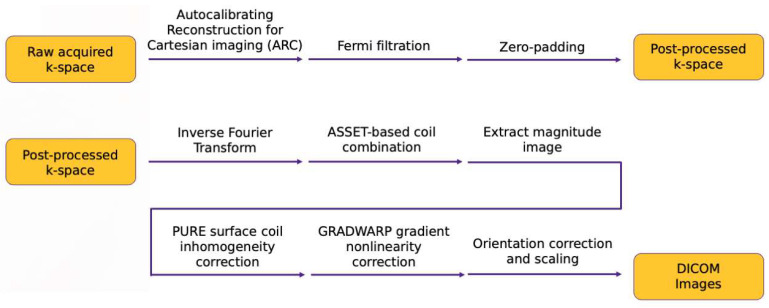
k-Space and image space post-processing steps for the in-house pipeline to reconstruct DICOM images from raw scanner data. Briefly, the steps in k-space are as follows: ARC reconstruction (parallel imaging), Fermi filtration to remove Gibbs artifacts, and zero-padding to bring the image to the intended output resolution. Image-space processing included coil combination, surface coil intensity correction, and gradient coil inhomogeneity correction.

**Figure 3 bioengineering-10-00267-f003:**
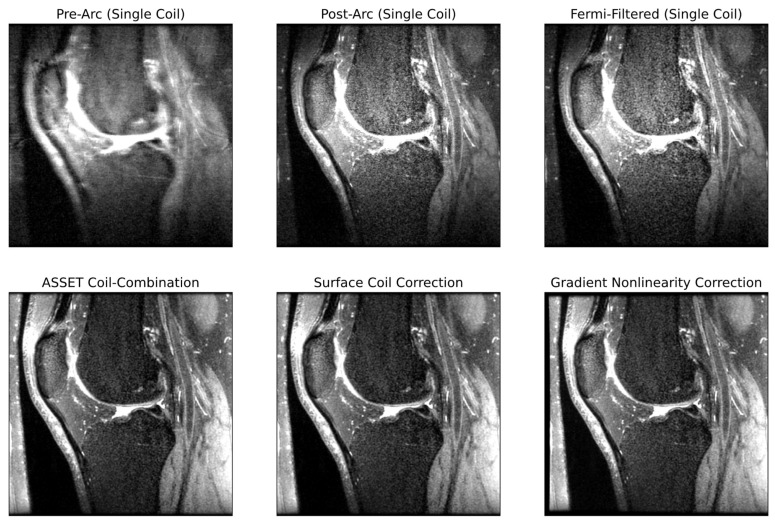
Intermediate outputs within the post-processing pipeline going from raw k-space to DICOM images. Each pane of the image reflects the output of the image after the step described by the pane title.

**Figure 4 bioengineering-10-00267-f004:**
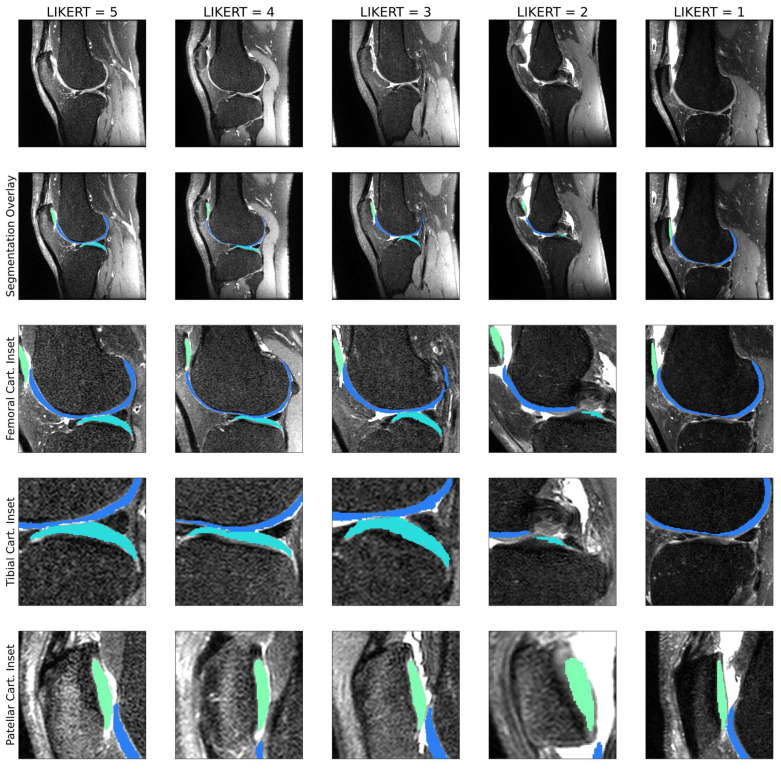
1–5 LIKERT cartilage segmentation scores overlaid on ground truth knee scans. In this example, the LIKERT of 5 indicates human-like segmentation; the LIKERT of 4 shows a slight underestimation of patellar and tibial cartilage; the LIKERT of 3 is assigned due to minor underestimation of patellar and tibial cartilage, with soft tissue detected as femoral cartilage; the LIKERT of 2 is assigned due to missing mask areas for patellar and tibial cartilage, with femoral cartilage overestimation; the LIKERT of 1 is missing a tibial cartilage mask.

**Figure 5 bioengineering-10-00267-f005:**
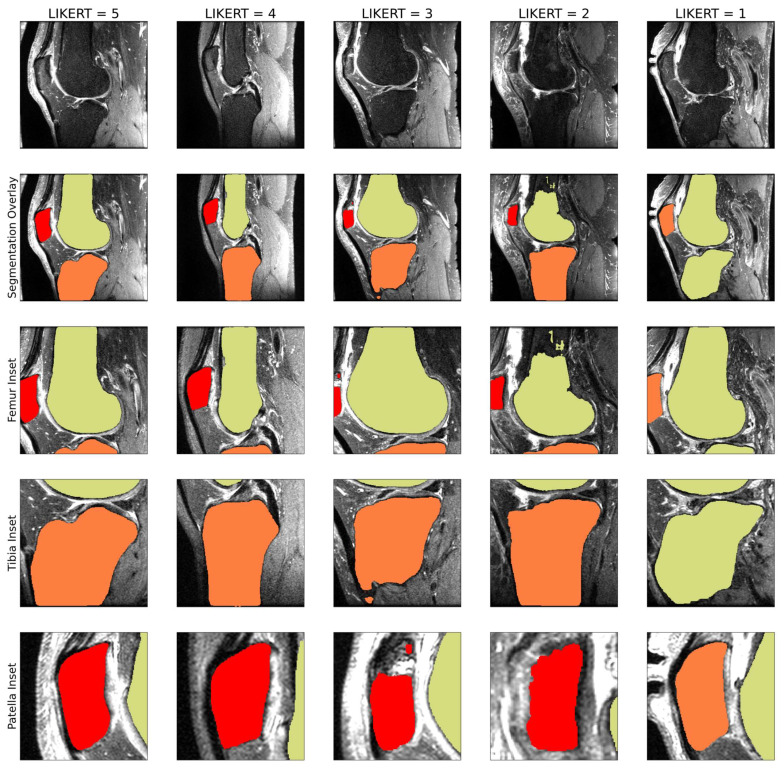
1–5 LIKERT bone segmentation scores overlaid on ground truth knee scans. In this example, the LIKERT of 5 indicates human-like segmentation; the LIKERT of 4 shows minor missing components in the femoral bone; the LIKERT of 3 shows missing components of the patellar bone mask; the LIKERT of 2 shows major missed regions within the tibial and patellar bone; the LIKERT of 1 has patella and tibia masks misassigned.

**Figure 6 bioengineering-10-00267-f006:**
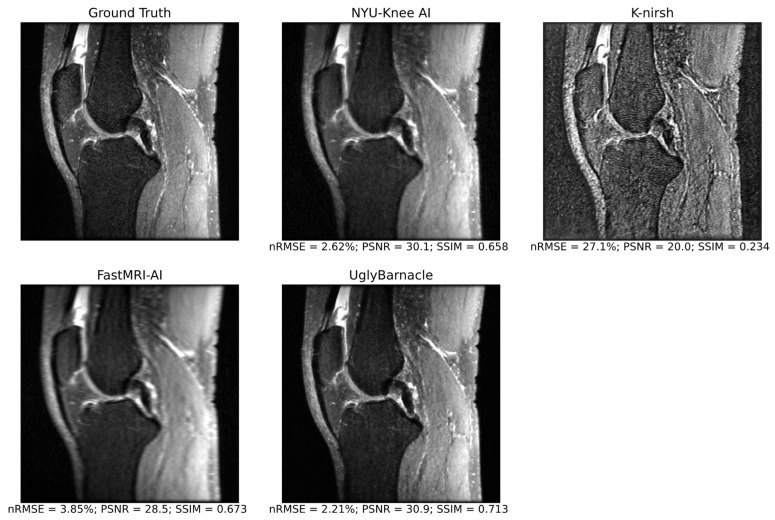
Intermediate pipeline reconstruction outputs for each of the top 4 submissions in an example sagittal slice, as well as ground truth, with reconstruction metrics displayed for the volume including the visualized slice. For this volume, UglyBarnacle delivers the highest quality reconstruction, followed closely by NYU-Knee AI, recovering sharpness and many fine details lost to aliasing during 8× Poisson undersampling. K-nirsh delivers an intermediate reconstruction that was poor by standard reconstruction metrics, but perceptually, made boundaries between tissues much more distinct and perhaps easier to segment. This is likely due to K-nirsh fine-tuning the reconstruction and segmentation networks in an end-to-end manner, unlike other top submissions.

**Figure 7 bioengineering-10-00267-f007:**
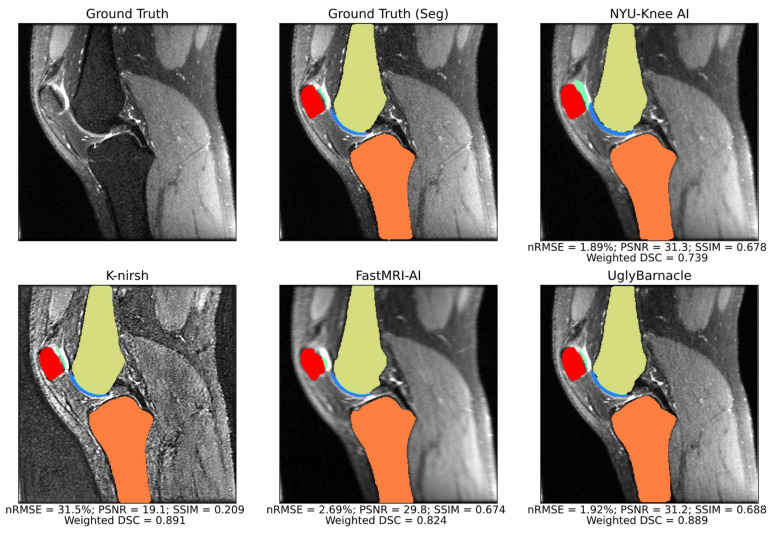
Sagittal slice segmentations overlaid on intermediate pipeline reconstructions, with reconstruction and segmentation metrics for the volume including the slice displayed. Background anatomy slices were thus blurrier for some teams than for others, as different teams had different quality intermediate pipeline reconstruction outputs. In this example, segmentation quality was strong for all top submissions, with only some overestimation of cartilage thickness from the NYU-Knee AI pipeline being apparent. K-nirsh maintains a slight edge over UglyBarnacle in reconstruction metrics for this volume.

**Figure 8 bioengineering-10-00267-f008:**
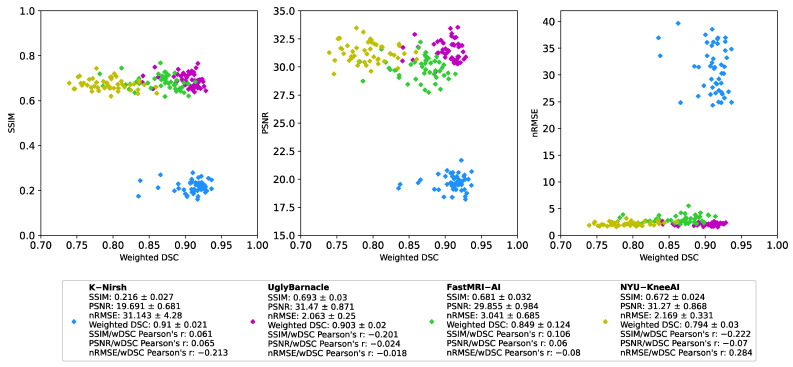
Reconstruction metrics (nRMSE, PSNR, SSIM) plotted against weighted DSC for each of the top four submissions, with each point denoting a subject in the test set (n = 50). Pearson’s correlation coefficient was calculated for each pair and is displayed on the chart, indicating that at absolute best, there was a weak correlation between segmentation and reconstruction metrics, and that in most cases, there was no or even negative correlation.

**Figure 9 bioengineering-10-00267-f009:**
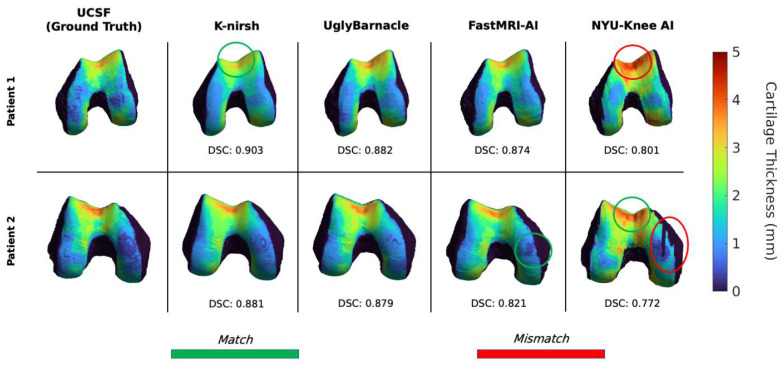
Femoral cartilage thickness maps projected onto voxel-based femoral bone shapes for each of the top 4 teams, as well as ground truth. While all submissions exhibit a degree of smoothness that is not reflected in the ground truth, the top three especially were strong in preserving cartilage thicknesses (K-nirsh, UglyBarnacle, FastMRI-AI), with NYU-Knee AI slightly overestimating cartilage thicknesses but still preserving key features in some regions.

**Figure 10 bioengineering-10-00267-f010:**
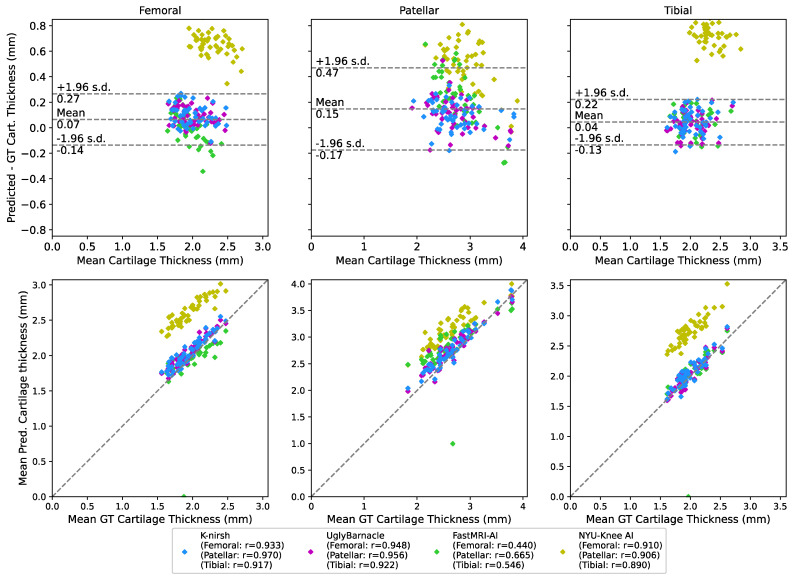
Bland-Altman and correlation plots between predicted and ground truth cartilage thicknesses for each of the top 4 submissions, across each of the 3 cartilage compartments. The mean and standard deviations for these plots were calculated using the data points from K-nirsh, UglyBarnacle, and FastMRI-AI, given the thickness overestimations seen from NYU-Knee AI. The top three submissions saw minimal bias and strong fidelity to ground truth, while NYU-Knee AI appeared to slightly overestimate particularly tibial and femoral cartilage thicknesses. That said, correlation plots showed strong correlations between predicted and ground truth thicknesses for K-nirsh, UglyBarnacle, and NYU-Knee AI. FastMRI-AI visually appeared to have strong correlation as well, but an outlier case appears to have severely degraded the correlation coefficient. All told, these results collectively are quite promising that submissions are suitable for some downstream biomarker analysis.

**Figure 11 bioengineering-10-00267-f011:**
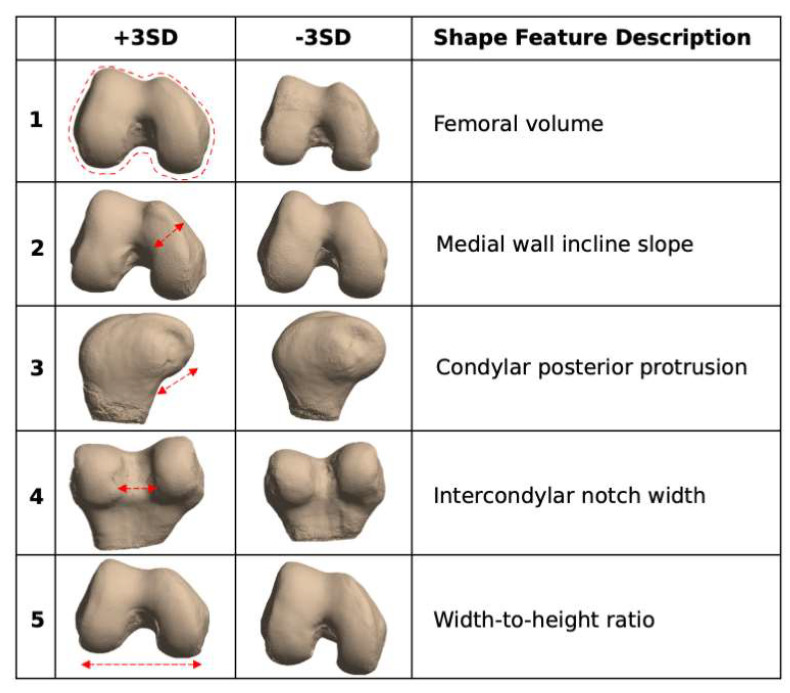
Femoral bone shape features, visualized after statistical parametrization, with qualitative descriptions of shape features. Similar features were also generated for the tibia and patella by the same procedure: extracting Euclidean points of bone surfaces, converting them into 1D vectors, using PCA to compress the resulting matrix into a 5-dimensional one, and visualizing each of the PCs.

**Figure 12 bioengineering-10-00267-f012:**
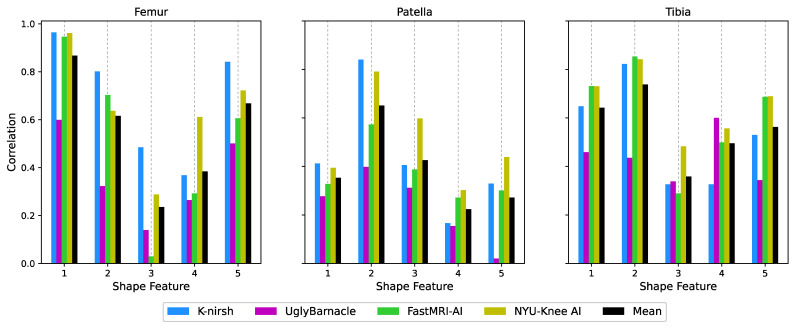
Correlations along femoral, tibial, and patellar bone shape features between submissions and ground truth. For many of the bone shape features, correlations were moderate to strong, indicating another means in which submitted segmentations from 8× undersampled images at times were suitable for downstream biomarker analysis. K-nirsh and NYU-Knee AI appeared to have strong correlations most consistently between predicted and ground truth bone shapes among the top 4 submissions.

**Table 1 bioengineering-10-00267-t001:** Acquisition parameters for 3D fat-suppressed CUBE sequence used in K2S dataset, and for this challenge.

MR Acquisition Information		
Scanner: GE Discovery MR750 3T Scanner (GE Healthcare, Milwaukee, WI) Gradient System Max Strength: 50 mT/m Max Slew Rate: 200 mT/m/ms		
Coil: 18-channel knee transmit/receive coil (Quality Electrodynamics (QED), Mayfield Village, OH)		
TR/TE: 1002/29 ms	FOV: 150 mm	Slice Thickness: 0.6 mm (0.6 mm spacing between slices)
Flip Angle: 90	SAR: 0.0939	Echo Train Length: 36
Frequency: 128	Bandwidth: 244	ARC [45]: 4 (R = 2 in k_y_, k_z_)
Acquisition Matrix: 256 × 256 × 200	Image Dimensions: 512 × 512 × 200 Resolution: 0.586 mm × 0.586 mm × 0.6 mm Voxel Size: 0.293 mm × 0.293 mm × 0.6 mm	

**Table 2 bioengineering-10-00267-t002:** Segmentation performance across test set (n = 50) for each of the top 4 pipelines, stratified by tissue compartment. Results are presented mean ± 1 s.d. K-nirsh showed the strongest results in each tissue compartment and overall, and is shown in bold.

	Cartilage			Bone			Full
Team	Femoral	Tibial	Patellar	Femur	Tibia	Patella	Weighted DSC
K-nirsh	0.904 ± 0.014	0.899 ± 0.015	0.910 ± 0.034	0.989 ± 0.002	0.985 ± 0.004	0.966 ± 0.012	0.910 ± 0.021
UglyBarnacle	0.895 ± 0.016	0.890 ± 0.017	0.903 ± 0.032	0.984 ± 0.004	0.980 ± 0.004	0.961 ± 0.015	0.903 ± 0.021
FastMRI-AI	0.845 ± 0.124	0.862 ± 0.126	0.843 ± 0.124	0.964 ± 0.078	0.952 ± 0.138	0.834 ± 0.306	0.849 ± 0.123
NYU-Knee AI	0.798 ± 0.029	0.756 ± 0.04	0.796 ± 0.043	0.980 ± 0.004	0.975 ± 0.005	0.939 ± 0.014	0.795 ± 0.030

**Table 3 bioengineering-10-00267-t003:** Standard reconstruction metrics for intermediate pipeline outputs from all top submissions across the released test set (n = 50). Results are presented mean ± 1 s.d. The top pipeline by each of these metrics was UglyBarnacle, shown in bold.

Team	nRMSE	PSNR	SSIM
K-nirsh	31.2 ± 4.26	19.7 ± 0.68	0.217 ± 0.059
UglyBarnacle	2.07 ± 0.25	31.5 ± 0.87	0.693 ± 0.043
FastMRI-AI	3.05 ± 0.68	29.8 ± 0.99	0.681 ± 0.061
NYU-Knee AI	2.18 ± 0.33	31.3 ± 0.87	0.672 ± 0.029

## Data Availability

During the challenge, prospective participants were able to request access to the K2S dataset at this link. Dataset access is available from the coauthors of the challenge upon reasonable request.

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
