# Peer review of "K2S Challenge: From Undersampled K-Space to Automatic Segmentation"

_bioengineering, 2023, doi:10.3390/bioengineering10020267_

Round 1
Reviewer 1 Report
This paper presents the setup, organization, and results of a challenge for joint reconstruction and segmentation of accelerated knee MRI data. A large dataset of raw scans was acquired and post-processed. A subset of the data was extensively manually segmented, and the remaining data was further semi-automatically segmented to arrive at a dataset ready for hosting a K2S-challenge at the MICCAI conference. Out of the participants, four top submissions were selected and assessed in full detail on application-specific metrics.
The paper is well-written and presents timely research. It advances in presenting an MRI reconstruction challenge beyond the state-of-the-art, by including the application domain in the evaluation. It thus leaves a platform for potential future use by methods yet to be developed in the field, thereby providing a solid benchmark.
The design choices are carefully motivated and discussed in the context of the clinical and practical setting, e.g., having accelerated ground truth data, and partly semi-automated segmentation.
Please find my comments below in sequential order.
l152 ‘its crisp resolution and relatively thin slices’ -> this may be rephrased by stating the advantage of 3D imaging relative to 2D imaging, yielding increased SNR allowing for higher resolution imaging, or similar.
Table 1: Is the FOV 150mm isotropic? With the acquisition matrix, image dimensions and resolution, the numbers do not seem to match. Please correct.
The images in Fig. 3 are dark and difficult to interpret on a printout.
Figs. 4 and 5 are small and the differences are difficult to read by this reviewer. Can insets be used, and pointers added, to highlight relevant structures? Likewise, Fig. 7 appears to suffer from mild jpeg-compression in the current version.
Table 2: Can differences between the top 2 teams be tested for significance? The scores are approximately half a standard deviation apart. With the power of n=50, this may reach significance.
Fig. 6: The K-nirsh appears to partly high-pass filter the input image. This is an interesting approach to this multi-domain problem. If the authors agree, could this be made more explicit in the discussion?
Figs. 8 & 10: please adjust the team order in the legend, and add larger color labels.
Fig. 10. The Bland-Altman analysis cannot be performed on joint results of all teams, given the large error for NYU-Knee AI. Please adjust and correct the analysis.
Minor: The results of the winning team are barely visible.
Is there a trend of a more negligible bias for larger thickness in the Femoral bone?
The oversegmentation of most methods(?) is currently visually not detectable in the images. Is this a matter of sensitivity in annotation? In any case, the effect is small, 0.1mm equals 50% of the reconstructed voxel size and 25% of the acquired voxel size, if I am correct.
Discussion
l.546 Here the 3D nature of the data may be emphasized.
l.557 ‘which likely played a substantial role…’. This is unknown. An ablation study with separate sequential reconstruction and segmentation of this method would be needed to verify and strengthen this claim. Given that this is the only end-to-end method, such an analysis is highly recommended.
l.574-l.576. A Bland-Altman analysis is more appropriate than a correlation analysis. Can the bias be compared between the top 2 methods? Is this significant? Or can individual exemplar scans be shown where an advantage for either method is clear?
l.608 and further: Can the authors motivate their choice for equidistant undersampling, perhaps scanner implementation constraints? It is well known that random undersampling yields better aliasing noise distribution and allows for higher acceleration factors. This may be stated as a limitation in the discussion section, by putting this challenge in the context of e.g. [Beauferris, Front Neuroscience 2022] where randomly undersampled 3D data were provided.
Reviewer 2 Report
The paper is well written and very detailed presentation of data and results were provided. I only have one concern whether the authors were able to ask for clear permission (maybe as part of the competition rules?) from the top teams for their analyses to be published in the paper.
